# Adaptive radiation by waves of gene transfer leads to fine-scale resource partitioning in marine microbes

Jan-Hendrik Hehemann[1,†,*], Philip Arevalo[2,*], Manoshi S. Datta[3,*], Xiaoqian Yu[4], Christopher H. Corzett[1], Andreas Henschel[1,†], Sarah P. Preheim[1,†], Sonia Timberlake[5,†], Eric J. Alm[1,5,6] & Martin F. Polz[1]

Adaptive radiations are important drivers of niche filling, since they rapidly adapt a single clade of organisms to ecological opportunities. Although thought to be common for animals and plants, adaptive radiations have remained difficult to document for microbes in the wild. Here we describe a recent adaptive radiation leading to fine-scale ecophysiological differentiation in the degradation of an algal glycan in a clade of closely related marine bacteria. Horizontal gene transfer is the primary driver in the diversification of the pathway leading to several ecophysiologically differentiated *Vibrionaceae* populations adapted to different physical forms of alginate. Pathway architecture is predictive of function and ecology, underscoring that horizontal gene transfer without extensive regulatory changes can rapidly assemble fully functional pathways in microbes.

[1] Department of Civil and Environmental Engineering, Massachusetts Institute of Technology, Cambridge, Massachusetts 02139, USA. [2] Microbiology Graduate Program, Massachusetts Institute of Technology, Cambridge, Massachusetts 02139, USA. [3] Computational and Systems Biology Graduate Program, Massachusetts Institute of Technology, Cambridge, Massachusetts 02139, USA. [4] Department of Biology, Massachusetts Institute of Technology, Cambridge, Massachusetts 02139, USA. [5] Department of Biological Engineering, Massachusetts Institute of Technology, Cambridge, Massachusetts 02139, USA. [6] Broad Institute, Cambridge, Massachusetts 02139, USA. † Present addresses: MARUM Center for Marine Environmental Sciences, University of Bremen Max Planck Institute for Marine Microbiology, Celsiusstraße 1, 28359 Bremen, Germany (J.-H.H.); Department of Electrical Engineering and Computer Science/Institute Center Smart Infrastructure (iSmart), Masdar Institute, 54224 Abu Dhabi, United Arab Emirates (A.H.); Department of Geography and Environmental Engineering, Johns Hopkins University, Baltimore, Maryland 21218, USA (S.P.P.); AbVitro, Inc. 27 Drydock Ave, Boston, Massachusetts, USA (S.T.). * These authors contributed equally to this work. Correspondence and requests for materials should be addressed to M.F.P. (email: mpolz@mit.edu).

Adaptive radiations are thought to have played an important role in the diversification of life. They manifest as rapid ecological differentiation of a single clade of organisms in response to ecological opportunity thought to arise primarily from resource availability following extinctions or colonization of new habitats[1,2]. A prime example are Darwin's finches, which quickly diverged from a single ancestor into several, locally adapted species on the Galapagos Islands due to evolvability of beak shape, which allowed rapid adaptation to novel resources. In recent years, laboratory evolution and mesocosm studies using microbes have provided experimental support for ecological opportunity triggering adaptive radiations[3–6]. However, for microbes in the wild, adaptive radiations, both ancient and recent, have remained difficult to document. First, the details of ancient diversifications are nearly impossible to reconstruct, since past ecological opportunities are often unknowable, and horizontal gene transfer (HGT) can erode phylogenetic signal. Furthermore, it is even questionable whether adaptive radiations might be possible in contemporary environments, considering that the long co-evolutionary history of microbes and their resources has led to high niche filling[6]. As a consequence, we do not know genetic mechanisms and ecological opportunities that could give rise to adaptive radiations in complex natural environments.

Here we ask whether a group of very closely related but ecologically differentially associated bacterial populations show the characteristics of an adaptive radiation, including rapid diversification of a single clade into multiple, ecologically differentiated clades, associated with traits adaptive towards environmental opportunities[1]. These populations were originally identified as genotypic clusters in protein-coding marker genes with differential distribution among size fractions within the same water samples indicating association with different resource types such as dissolved or particulate organic matter and zoo- or phytoplankton[7–10]. Subsequent work has established that these clusters also act as gene flow[11], social[12] and behavioural[13] units suggesting that they possess many attributes commonly associated with sexual species. However, because our sampling scheme considers only bacteria co-existing in small-scale microhabitats, we designate them as populations to which we assign species names if a previously described type strain falls within the genotypic cluster identified as a distinct population.

Our test case is a clade of very closely related Vibrionaceae isolates, which we previously hypothesized to comprise at least seven recently speciated populations based on their genetic and environmental structure. We show that this clade rapidly diversified into population-specific ecophysiological types specialized for the degradation of different physical manifestations (chain length, solubility and concentration) of the same algal glycan. This specialization is manifest as unique pathway configurations that arose by extensive horizontal gene transfer and are highly predictive of metabolic performance. We first reconstruct the evolution of the different pathway types and characterize their physiological properties. We then show that environmental associations are consistent with the physiological predictions and propose a model of glycan degradation involving the evolution of interacting populations.

## Results

### Alginate degradation pathways differentiate populations.
We first asked whether adaptive changes can be hypothesized based on comparison of 84 genomes representing the Vibrionaceae populations, including a clade of seven very recently diverged populations (crown group) (Fig. 1, grey box). This analysis highlighted a pathway specific for the degradation of the brown algal glycan alginate as having undergone extensive evolutionary changes across the majority of populations (Fig. 1a,b; Supplementary Fig. 1). These include both population-specific presence and absence of the pathway, as well as major differences in its architecture. For example, several populations contain a canonical pathway consisting of four polysaccharide lyase (PL) families[14], while others lack up to three of the four lyase families (Fig. 1b). These four families perform different molecular functions in the alginate pathway: alginate lyases (Aly) PL6 and PL7 initiate the extracellular lysis of the polymer, and members of two oligoalginate lyase (Oal) families PL15 and PL17 complete degradation into monomers[15–18]. The latter two gene families are the keystone carbohydrate active enzymes[19] of the core pathway, since they generate sugar monomers that can be further catabolized, and their absence abolishes pathway functionality. These initial observations suggested the possibility of a fine-grained analysis of the evolutionary history and potential adaptive significance of the alginate pathway differentiation.

### Pathways have assembled primarily horizontally.
Cursory inspection of the alginate pathway across our Vibrionaceae populations appears consistent with an ancient, single HGT since a core set of alginate degradation genes is present in a majority of the clade including the deeply branching Aliivibrio (Fig. 1a; Supplementary Figs 1 and 2). However, detailed phylogenetic reconstruction (Methods) that includes an additional 395 high-quality genomes obtained from Genbank for reference (Supplementary Fig. 3) reveals an unexpectedly complex history (Fig. 1d,e). In most cases, multiple copies of each alginate lyase family represent independently evolving subfamilies that did not arise by duplication within the Vibrionaceae (see Methods for statistical support for definition of subfamiles). In fact, there is little vertical descent and the majority of clades with alginate degradation pathways acquired both Oal and Aly genes horizontally (Fig. 1d,e). Across all our populations, transfer of Oal genes was so common that every population exchanged at least one gene copy with at least one other population (Fig. 1d). Even among the seven closely related populations of the crown group we estimate three independent initial acquisitions of Oal subfamilies from a variety of sources followed by lateral spread among populations and acquisitions of new subfamilies (Fig. 1a,d). Moreover, Alys and Oals are distributed across multiple regions on chromosome 1, chromosome 2 and a putative extrachromosomal element in one Vibrio breoganii (FF50) and one Vibrio sp. F13 (9CS106) isolate with nearly closed genomes. These regions are significantly enriched in genes annotated as mobile elements, transposases and integrases (hypergeometric test, $P = 0.0019$). Some of these regions also display significantly decreased GC content consistent with the recent introduction of foreign DNA (Supplementary Table 1). Hence multiple lines of evidence reject the seemingly ancient acquisition and subsequent vertical modification of the core pathway and instead suggest multiple recent acquisitions and transfers.

The core pathway of Oals was extended in a surprisingly rapid and complex sequence of events by independent acquisitions and transfers of Aly families PL6 and PL7. Similar to the Oals, Aly genes also spread extensively within the crown group by independent acquisition and transfer. However, these genes were lost in a lineage-specific manner within Vibrio tasmaniensis and Vibrio lentus (Fig. 1d,e). Furthermore, Vibrio sp. F10 never acquired Aly genes despite possessing both Oals (Fig. 1d,e). The more basal Vibrio groups, V. breoganii, Vibrio rumoiensis and the Aliivibrio, all independently acquired these genes as well and transferred a number to the crown group (Fig. 1e). Taken together, several different pathways were assembled by an

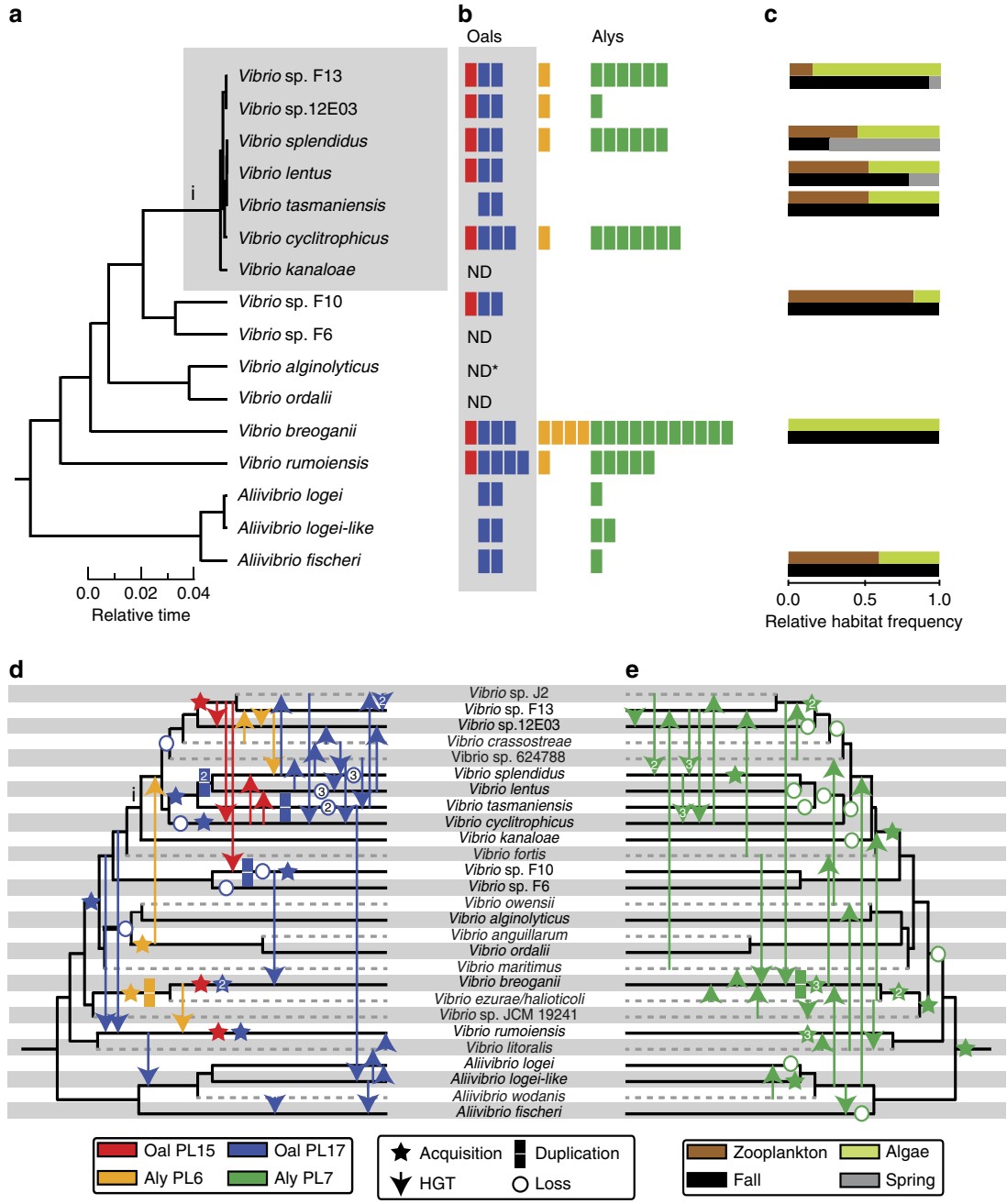

**Figure 1 | Evolutionary history and ecological occurrence of alginate lyases. (a)** Relative timed maximum-likelihood phylogeny of *Vibrionaceae* populations co-occurring in the same water samples. Species names are assigned if a previously described type strain falls within the population; otherwise, the designation *Vibrio* sp. is given. **(b)** Maximum copy number of alginate lyase families within members of a population identified by presence of a single enzymatic domain are represented by coloured rectangles. ND indicates that Alys and Olys were not detected. ND* indicates that while no Alys or Olys were detected in *V. alginolyticus* isolates from our collection, it is present in other *V. alginolyticus* isolates from geographically distant locations (Supplementary Fig. 1). **(c)** Normalized distribution of isolates obtained from algal detritus particles and zooplankton handpicked under a dissecting microscope and phylogenetically categorized by multilocus gene analysis for two seasonal samples. **(d,e)** Phylogenetic reconciliation (Methods) by comparison of pathway-specific gene trees (Supplementary Figs 6–9) and a timed 'species' tree showing the history of each of four lyase gene families embedded in the reference species phylogeny: **(d)** Oal domains PL15 and PL17, and Aly domain PL6; **(e)** Aly domain PL7. Acquisition represents an independent entry of a subfamily into a clade within our collection. Solid and dashed lines on the phylogenetic tree indicate clades represented in our collection or obtained from Genbank, respectively. Numbers within symbols indicate multiple independent occurrences of the represented event. Within-population HGT and duplication are not depicted. Lowercase Roman numeral i indicates the crown group consisting of seven closely related populations.

evolutionary Ping Pong of rapid back and forth transfers among lineages; the pace of this is evident in the crown group of vibrios that are nearly indistinguishable in ribosomal protein gene sequences yet contain populations that have lost the pathway or acquired a range of Alys in addition to the core set of Oal genes (Fig. 1; Supplementary Fig. 1).

The considerable variation in Aly gene copy number, which arose primarily by acquisition and loss rather than duplication,

differentiates pathways functionally, highlighting the key role of HGT and the flexible genome in niche differentiation among populations. In particular, while PL6 and PL7 gene copies are absent in several lineages, they are especially abundant in *V. breoganii*, *Vibrio cyclitrophicus*, *Vibrio splendidus* and *Vibrio* sp. F13. In *V. breoganii*, the pathway underwent the most significant expansion, involving duplications of PL6 and PL7 genes in addition to several PL7 gene transfers from multiple sources (Fig. 1d,e). Within each population, further variation in PL7 copy number also exists (Supplementary Figs 1 and 4) but it is unclear whether such variation is due primarily to transfer, loss or duplication. Regardless, these observed differences in gene copy number have important physiological consequences; populations possessing more Aly gene copies showed increased enzyme expression (Supplementary Fig. 5) and enzymatic activity when exposed to alginate (Fig. 2). Because many of the genes are distributed across different regions in the genome and hence not co-regulated, increased activity rapidly evolved, in large part by gene acquisition and expression. The gene acquisition we observed is reminiscent of molecular cloning and exemplifies that the process works in the laboratory because microbes are well adapted for incorporation and expression of heterologous genes.

**Populations possess different ecophysiological strategies.** We next asked how differences in pathway architecture might shape ecological niches at the population level. We first hypothesized that populations possessing only Oal genes may only have limited ability to utilize oligoalginate molecules. We therefore performed growth experiments on alginate of high (degree of polymerization, Dp > 50), medium (Dp ∼ 20), and low (Dp ∼ 3–4) molecular weight, which reflects the potential resource space, since extracellular glycan depolymerization relatively inefficiently retains breakdown products of different size, hence liberating oligoalginate molecules for consumption by other microbes[20]. This showed that, among 55 surveyed *Vibrionaceae* strains, the presence of at least one Aly and one Oal family (as in *V. cyclitrophicus*, *V. breoganii*, *V. splendidus* and *Vibrio* sp. F13) perfectly predicted the ability to grow on polymeric forms of

alginate (Dp > 50 and ∼ 20; Fisher's exact test, $P = 4 \times 10^{-11}$). In contrast, possessing only Oal families (as in *V. tasmaniensis* and *Vibrio* sp. F10) conferred the ability to grow only on low-molecular weight oligomers (Dp ∼ 3–4; Fig. 3; Fisher's exact test, $P = 1 \times 10^{-9}$). Since these oligomers are Aly digestion products, populations that lack Alys (but still possess Oal families) may 'scavenge' substrates produced by Aly-possessing populations. Our findings extend similar interactions recently suggested among bacteria in the gut[21,22] to ocean microbial communities, which persist in a much more dilute environment suggesting the observed differentiation represents a general principle of glycan degradation.

Populations are differentiated along an additional niche axis: the speed with which they can access the intact alginate polymer due to differential solubility of Alys. We noticed that although most Aly-possessing populations had very short lag phases on high-molecular weight polymer (Dp > 50) (Fig. 4a, for example, 13B01 and 12B01), a subset of isolates displayed long lag phases – over 24 h in some cases – that increased with polymer length (Fig. 4a, for example, 1F157 and ZF211). We hypothesized that short lag phases are enabled by broadcasting Alys into the three-dimensional polymer matrix while long lag phases occur in isolates that tether the enzymes to the cell allowing only access to the two-dimensional polymer surface. Consistent with this hypothesis, broadcast alginate lyase activity was high among isolates with short lag phases (Fig. 4b–d, for example, 13B01 and 12B01), but comparatively low in isolates with longer lag phases (Fig. 4b–d, for example, 1F157 and ZF211). Notably, membrane-bound and intracellular alginate lyase activity was comparable among isolates, regardless of lag phase length (Fig. 4b,c). Furthermore, in a plate-based assay intended to visualize broadcasted alginate lyase activity, isolates with short lag phases created large halos of lyase activity that extended far beyond the colony boundary (Fig. 4d). These halos were small or absent for isolates displaying long lag phase. Interestingly, this long lag phase phenotype arose independently as a population-level characteristic among all *Aliivibrio fischeri* isolates capable of degrading alginate and a within-population polymorphism in *V. splendidus* displayed by ∼ 5% of isolates. Our results also provide an explanation for recent observation of lag phases among some *Alteromonas macleodii* isolates[23] and suggest this phenotype might be common.

Our plate-based enzymatic broadcasting assay also revealed additional polymorphisms in the strength of the broadcasting phenotype. Some isolates appeared to be 'super broadcasters' with unusually large halos, indicating their superior ability to degrade high-molecular weight alginate (Fig. 4d). This phenotype can be traced back to the acquisition of a PL7 in a subset of *V. splendidus*, including strain 13B01. Interestingly, a super broadcaster phenotype was recently bioengineered for production of bioethanol from algal biomass by combining the alginate pathway of *V. splendidus* 12B01, a low broadcaster, with an engineered PL7 enzyme that is secreted by *Escherichia coli*[15]. Hence our analysis demonstrates that nature found an identical solution to the problem of rapid access of the insoluble polymer and highlights fine-scale physiological differences as a resource for bioengineering. Ecologically, isolates that broadcast enzymes may act as 'pioneers', which have a competitive advantage when colonizing native substrates. By contrast, while isolates with long lag phases possess the full repertoire of alginate lyases (both Alys and Oals), they likely grow more rapidly in the presence of isolates that broadcast alginate lyases into the environment. We therefore refer to them as 'harvesters', since they may harvest the fruits of enzymes sown by pioneers.

The observed physiological variation suggests differential adaptation of the populations to alginate and its partial

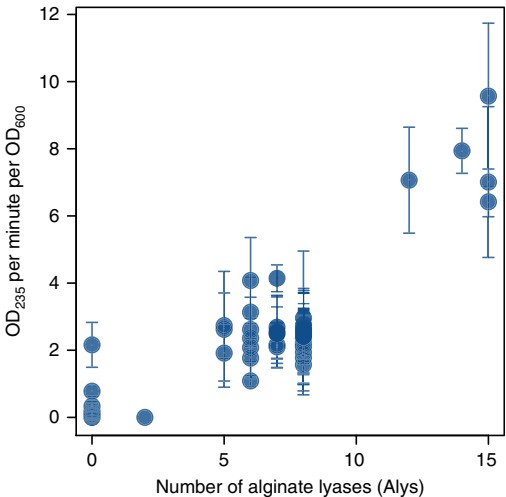

**Figure 2 | Alginate lyase activity is modulated by gene dosage.** Isolates were grown in marine broth with added alginate oligosaccharides to induce enzyme expression. The cells were lysed to determine the total, cell associated alginate lyase activity by measuring the increase of absorption at 235 nm with alginate as enzyme substrate. The alginate lyase activity of each strain was normalized against the optical density of the respective cell culture measured at 600 nm. The experiment was carried out in triplicate and the error bars display the standard deviation of the mean.

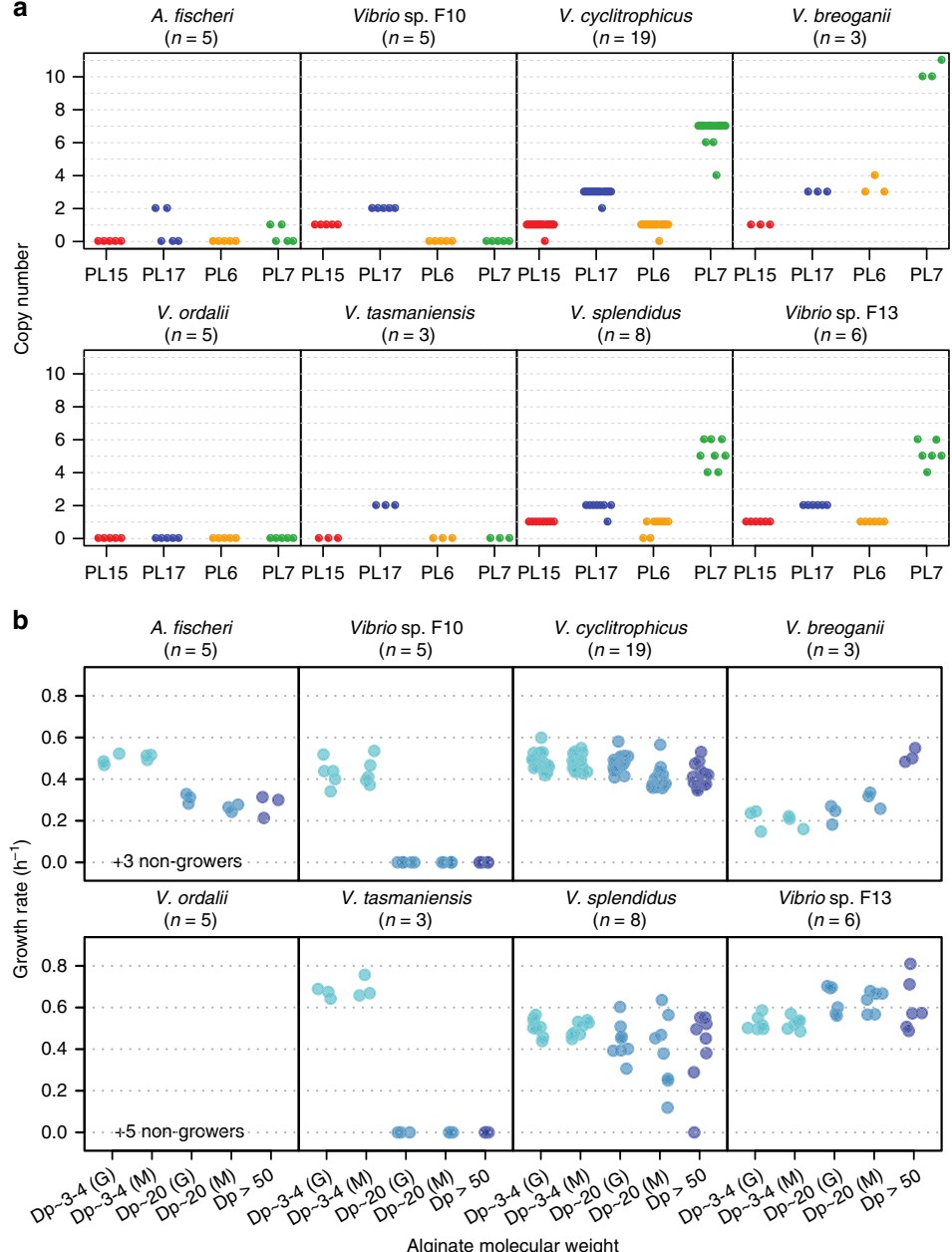

**Figure 3 | Growth rates on alginate substrates of different chain length and solubility.** (**a**) Alginate lyase (PL6, orange; PL7, green) and oligoalginate lyase (PL15, red; PL17, blue) copy number for individual strains within populations of *Vibrionaceae*. (**b**) Isolates representing different populations were grown in seawater minimal medium containing low- (Dp ~ 3–4, aqua), medium- (Dp ~ 20, blue), or high- (degree of polymerization, Dp > 50, dark purple) molecular weight alginate as the sole carbon source. Since alginate is a heteropolymer of guluronate and mannuronate, the low and medium molecular weight alginate was further purified into guluronate (G)- or mannuronate (M)-enriched fractions. Each dot represents the average growth rate of an isolate from the denoted population on the designated carbon source across three technical replicates. The number of isolates assayed per population (*n*) is indicated.

degradation products, which ultimately derive from algal cell walls. Yet *Vibrionaceae* are generally regarded to be animal (especially zooplankton) associated, including many facultative pathogens[24]. Hence, we tested to what extent possession of the alginate pathway is linked to association with dead algal biomass. To identify habitat association, we collected small particles recognizable under the microscope as algal detritus, and, for comparison, live and dead zooplankton (primarily copepods) during the fall and spring season, and obtained isolates on selective media[8] (Fig. 1c). Presence of Aly genes is strongly associated with presence on algal particles (Fisher's exact test,

$P = 1.68 \times 10^{-7}$). Furthermore, at least six populations (*A. fischeri*, *V. breoganii*, *Vibrio sp.* F13, *V. tasmaniensis*, *V. lentus* and *V. splendidus*) overlap in their habitat preferences by occurring on algal detritus, albeit *V. splendidus* was more strongly represented in spring, that is, cold water samples. Importantly, none of the populations that lack the entire alginate degradation pathway could be isolated from algal detritus. Moreover, several populations capable of alginate degradation, which were absent from algal biomass, might occupy different environmental microhabitats. For example, *V. cyclitrophicus* has previously been hypothesized to co-occur with unicellular algae in

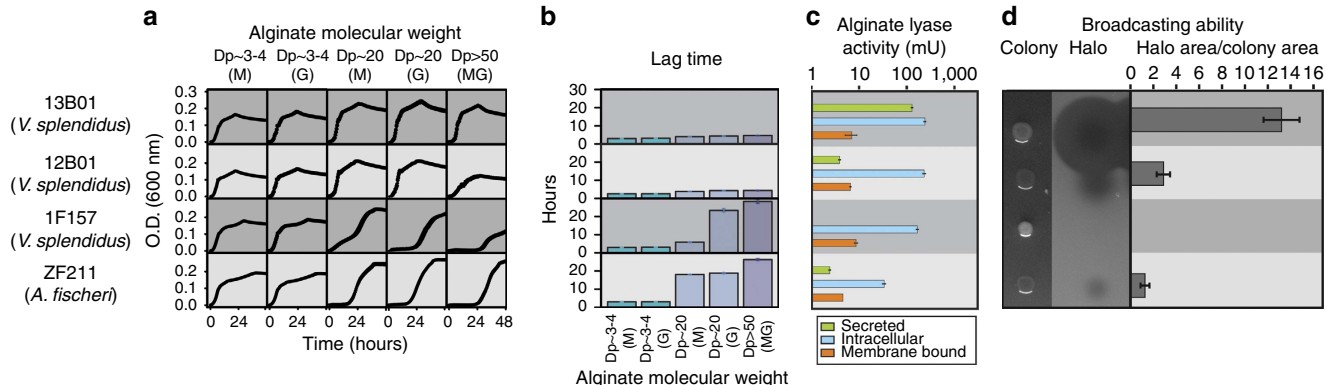

**Figure 4 | Membrane-bound versus broadcasted alginate lyases dictate growth lag time.** (**a**) Growth curves of isolates representing distinct pathway architectures on high- (degree of polymerization, Dp > 50), medium- (Dp ~ 20), or low- (Dp ~ 3-4) molecular weight alginate. Low and medium molecular weight alginate was further purified into mannuronate (M)- or guluronate (G)- enriched fractions. (**b**) Quantified lag time differences between strains. (**c**) The cellular milieu was fractionated into extracellular (secreted), membrane-bound, and intracellular components. For each fraction, alginate lyase activity was measured using a bulk enzymatic activity assay (Methods). Among isolates assayed, those with longer lag phases displayed reduced broadcasted alginate lyase activity, despite similar levels of intracellular- and membrane-bound alginate lyase activity. Bar diagrams represent averaged technical replicates ($n = 3$). Error bars represent standard deviations of the mean. One unit of activity defines an increase of 1.0 in absorbance at 235 nm per min. (**d**) Broadcasted alginate lyase activity measured independently with a plate-based assay (Methods). The size of the halo indicates the degree of broadcasted alginate lyase activity after a fixed period of time. Bar diagrams represent means of technical replicates ($n = 5$) and error bars represent s.d.

the water column[10] while *Vibrio* sp. F10, although present on algal biomass, is primarily associated with zooplankton (Fig. 1c)[8]. Hence several populations may spatially and temporarily partition the alginate substrate. However, at least four populations (*V. breoganii* and *Vibrio* sp. F13, *A. fischeri* and *V. tasmaniensis*) have high potential for interaction on algal detritus.

## Discussion

Combining phylogenetic, physiological, and environmental data, we suggest that a horizontally acquired alginate degradation pathway has undergone an adaptive radiation, which may mitigate competitive exclusion and enable a degradation cascade involving three ecophysiological types. First, 'pioneers' – alginate degraders with Oals and broadcasted Alys – colonize and degrade the intact polymer, thereby creating more soluble forms of the polymer and oligomers (Fig. 5). In the process, pioneers construct a niche for two other types of populations, which we refer to as 'scavengers' and 'harvesters'. Scavengers, which only have Oals, are 'cheaters' that cannot degrade alginate directly, but can take advantage of small oligomers (Dp ~ 3–4) produced extracellularly by the pioneers. By contrast, harvesters represent an intermediate between the pioneers and scavengers. Like pioneers, harvesters possess both Oals and Alys. However, instead of broadcasting the Aly enzymes, harvesters tether Alys to their cell surface. Furthermore, like scavengers, harvesters may also take advantage of small alginate oligomers produced by pioneers.

In theory, there are many mechanisms that might support the coexistence of these three ecological strategies in nature. For pioneers and scavengers, these include spatial structure[25,26] and asymmetric access to nutrients[27]. For harvesters, their lack of broadcast enzymes leads to a growth detriment (through long lag phases during growth on high-molecular weight alginate), but also makes them less likely to share their enzymatic degradation products. Thus, harvester populations may not be as prone to invasion by scavengers – as has been recently described for select human gut Bacteroidetes[28] – thereby allowing them to coexist with pioneers and scavengers. Finally, even different pioneer populations (*V. breoganii* and *Vibrio* sp. F13) are further ecologically differentiated by enzymatic activity levels, stemming from distinct pathway architectures (Fig. 2; Supplementary

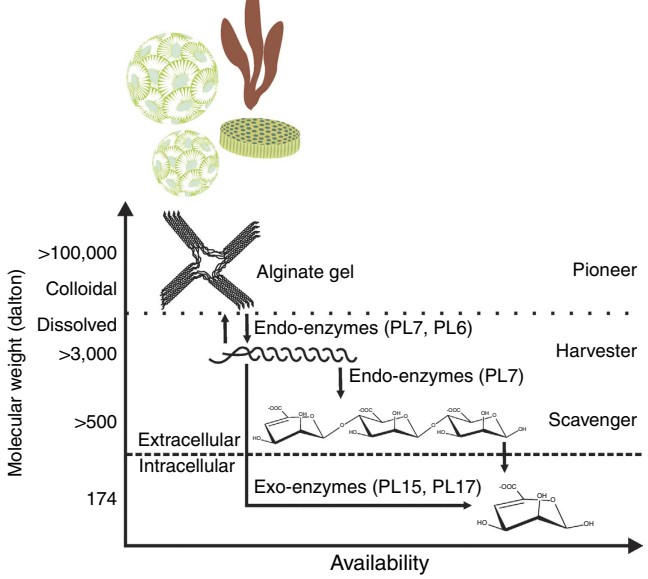

**Figure 5 | Alginate degradation cascade of substrates varying in solubility and chain length.** Marine vibrios diversified into different populations characterized by their ability to consume insoluble alginate polysaccharide and soluble alginate oligosaccharides of different chain lengths. Pioneers are specialized in consumption of native, insoluble alginate due to their endowment with broadcast alginate lyases. These enzymes can diffuse freely into the alginate gel and depolymerize the alginate into soluble oligosaccharides. Harvester populations with secreted but tethered alginate lyases can exploit the range of soluble alginate substrates including medium and small oligosaccharides liberated by pioneers. Scavenger populations devoid of any alginate lyases can only use the smallest alginate oligosaccharides.

Fig. 5), which may allow for their coexistence. Nonetheless, interactions may lead to fluctuations in populations and further work will be required to determine how stably pioneers, scavengers, and harvesters can coexist in the wild.

Our analysis shows that a very general ecological opportunity creates a surprisingly strong selective regime as evidenced by the

rapid, repeated evolution of different ecophysiological types among closely related bacteria. This finding underscores the general evolvability of microbes consistent with adaptive radiations documented in experimental evolution studies[3–5]. The mechanism, however, appears fundamentally different. HGT assembles highly nuanced functional pathways from different sources with apparent speed and facility. This includes increase in enzymatic activity, which appears driven by acquisition of gene copies, rather than changes in regulation. This importance of HGT is consistent with recent insights into the evolution of beak shape in the Galapagos Finches, which was accompanied by extensive interspecies gene flow[29] underscoring that in both bacteria and animals, gene import into the population rather than de novo evolution may be important during rapid diversification[30]. Finally, our results suggest that nuances in the architecture of the same pathway are predictable for ecological association and coexistence among diversifying populations, suggesting that such variation must be explored in detail if we are to understand the rules of microbial community assembly.

## Methods

**Isolates and culture conditions.** Strains tested here originated from previous studies on the ecological population structure of *Vibrionaceae*. Briefly, isolates were obtained either from size fractionated water samples, handpicked algal detritus particles and zooplankton, or different body parts of marine invertebrates by plating samples on *Vibrio*-selective marine TCBS media[7,8]. Individual colonies were picked and purified by re-streaking three times.

**Genome sequencing and assembly.** DNA from strains 1S159, 1S128, 1S175, 1S165, 5F7, FF227, ZF47, 5F23, 5F33, 5F97, ZF73, 5F306, ZF25 and 5F146 was extracted with the DNeasy Blood and Tissue Kit using the protocol for Gram-negative bacteria (Qiagen). Sequencing libraries were prepared using the Nextera DNA Library Preparation Kit (Illumina). Each strain was barcoded and sequenced in a $100 \times 100$ bp multiplex run on the HiSeq 2500 in Rapid mode (Illumina) at the Whitehead Institute Genome Technology Core (Cambridge, MA). Sequence reads were demultiplexed using a custom python script and imported into CLC Genomics Workbench 8.0.2 (http://www.clcbio.com) for further processing and assembly. Adaptors and low-quality regions were trimmed from the demultiplexed sequences and overlapping paired reads were merged. The final assemblies were performed using the CLC assembler with read mapping correction.

Draft genome assemblies were available for 63 strains indicated in Supplementary Table 2. Assemblies for some of these strains were refined by resequencing. DNA was isolated and short read sequencing libraries were prepared by random shearing per Illumina's protocol and sequenced short (51–71 bp) single-end reads on an Illumina GI, or paired-end reads on an Illumina GAII (ref. 11). Genome assemblies for strains 12B01 and 12G01 were corrected with these short read sequences. Additionally, long-insert libraries were prepared for 11 genomes representing different populations (1F53, ZF14, ZF29, 9ZC13, 12B09, 5S149, 5S101, FF33, 1F211, ZS17 and FF500) to aid in scaffolding assemblies. Final assemblies using these reference strains was performed by mapping short read sequences to the most closely related long-insert assembly as previously described[11]. This hybrid short read and long-insert scaffolding approach was also used to generate *de novo* assemblies for six additional strains (5F59, ZF57, FF273, 1A06, 9CSC122 and 13B01).

Strains 9CS106 and FF50 were sequenced using the PacBio RSII at the Yale Center for Genome Analysis. Initial assemblies were performed using the SMRT Portal Software at the MIT BioMicro Center and the HGAP Assembly #2 algorithm. When appropriate, assemblies were circularized with the minimus software package and assemblies were refined using the Resequencing #1 analysis in SMRT Portal. This assembly was corrected by mapping the short read Illumina sequences in CLC Genomics Workbench 8. This corrected assembly resulted in two closed chromosomes and one circular extrachromosomal element in FF50, and two nearly closed chromosomes and two circular extrachromosomal elements in 9CS106.

An additional 395 *Vibrionaceae* and 51 *Shewanella* genomes were retrieved from Genbank (ftp://ftp.ncbi.nlm.nih.gov/genomes/) for a total of 530 genomes analysed.

**Reference phylogeny construction.** Ribosomal proteins were identified using hidden Markov models (HMMs) constructed from a previously published alignment of bacterial and archaeal ribosomal proteins[31]. These were searched against all ORFs from the 84 sequenced genomes with hmmsearch (http://hmmer.org). ORFs matching ribosomal proteins with an e-value greater than $10^{-10}$ were excluded from further analysis. Paralogous ribosomal proteins were also excluded. All remaining ribosomal proteins present in at least 50% of all isolates were aligned

using the MAFFT-L-INS-i algorithm with default parameters[32]. Corresponding nucleotide sequences were aligned with the protein alignment as a guide using PyCogent[33]. A maximum-likelihood tree was constructed under the $GTR + G + F$ model of sequence evolution in RaxML[34]. The tree was rooted using *Shewanella* as an outgroup. A relative timed tree was created from the maximum-likelihood tree using RELTIME[35].

**Identification of alginate lyase domains.** HMMs of Alys PL6 and PL7 and Oals PL15 and PL17 were obtained from the dbCAN database[36]. These were searched against all ORFs from the 530 genomes using hmmsearch (http://hmmer.org). Best-scoring domains were then filtered by e-value ($<10^{-23}$) and alignment coverage ($>0.8$) both of which are more stringent cutoffs than those recommended by dbCAN (http://csbl.bmb.uga.edu/dbCAN/). These parameters were chosen to minimize false positives. The remaining domains for each lyase and the corresponding dbCAN domain set were aligned with MAFFT-L-INS-i using the dbCAN set as a seed under default parameters. Maximum-likelihood phylogenies were made from each of the four lyase domain alignments in the same manner as the reference phylogeny with 100 bootstrap replicates.

**Statistical support of alginate lyase subfamilies.** We initially defined subfamilies of each Aly and Oal as *Vibrionaceae*-specific clades that are divided by non-*Vibrionaceae* outgroups. Some of these clades were, however, so closely related that two alternative scenarios were possible: either horizontal acquisition of two different (albeit closely related) subfamilies or vertical evolution within the *Vibrionaceae* and transfer to the outgropus. We therefore tested the robustness of these initial subfamilies to arrive at the most conservative evolutionary scenario using the following methods: (1) determining bootstrap support values for *Vibrionaceae* clades and their non-*Vibrionaceae* outgroups; (2) testing the inferred ML topology against a more conservative topology (that is, a topology that grouped *Vibrionaceae* genes into a single clade) with the approximately unbiased test (AU test)[37]; (3) comparing branch lengths within a given subfamily phylogeny to branch lengths within a related subfamily with an origin supported by criteria 1 and 2. We performed these analyses on the PL17, PL6 and PL7 phylogenies. PL15 was not included in this analysis as only one PL15 subfamily was found.

For PL17, originally five subfamilies were inferred. However, bootstrap support for all of these subfamilies was low (all values $<60$). We tested the alternative hypothesis that subfamilies 1, 2 and 3 were a monophyletic group with the AU test. This topology was not significantly different from the inferred ML topology ($P = 0.339$). Performing the same test on subfamilies 4 and 5 yielded the same result ($P = 0.118$). However, the alternative topology with subfamilies 1, 2, 3, 4 and 5 as a monophyletic group was significantly different from the ML topology ($P = 7 \times 10^{-60}$) and we were able to reject that alternative hypothesis. Therefore we concluded that there are two independently evolving subfamilies of PL17.

For PL6, originally two subfamilies were inferred. Subfamily 1 was well-supported by bootstraps (97) while subfamily 2 was not (42). We tested the alternative hypothesis that subfamilies 1 and 2 were a monophyletic group and found that this topology was significantly different from the inferred ML topology ($P = 2 \times 10^{-6}$). Therefore we concluded that there are two independently evolving subfamilies of PL6.

For PL7, originally 14 subfamilies were inferred. Subfamilies 1 and 14 were supported by high bootstraps (100 and 90, respectively). Subfamily 13 was most closely related to subfamily 14 and therefore we tested the alternative hypothesis that these two subfamilies formed a monophyletic clade. This alternative topology was significantly different from the ML topology ($P = 0.017$) and therefore we were able to reject that alternative hypothesis.

Subfamilies 3–12 form a large clade along with a number of non-*Vibrio* members. Because of this, testing an alternative hypothesis was not appropriate given the number of possible alternatives available. We took advantage of the fact that subfamily 14 originated in the ancestor of *Aliivibrio* and *Vibrio*, making it the oldest of all *Vibrionaceae* PL7s tested. The longest root-to-leaf branch length in this subfamily is indicative of the maximum amount of divergence one would expect within any *Vibrionaceae*-specific PL7 subfamily. Using this value (0.66 substitutions per site) as a cutoff, we were able to recover subfamilies 3–8 with UPGMA clustering by phylogenetic distance (Supplementary Fig. 10). However, subfamilies 9–12 were merged into a single subfamily. The distance between subfamily 2 and all other subfamilies greatly exceeds the cutoff described above. Therefore we concluded that there are 11 independently evolving subfamilies of PL7.

The final tree with presence/absence information of lyases and each of the lyase gene trees were visualized using the Interactive Tree Of Life[38].

**Phylogenetic reconciliation.** The Aly and Oal gene trees were reconciled against the reference phylogeny using AnGST[39] with the following cost parameters: loss = 1.0, duplication = 2.0, HGT = 3.0 (ref. 40). The optimal reconciliation was the reconciliation with the lowest total event score. We also modified the AnGST source code to examine reconciliations with alternative gene birth scenarios that yielded the same minimal event cost. In all but one case (PL7 subfamily 2), the history of Alys and Olys within the set of 84 isolates from our collection was

unaffected by different birth scenarios. For this subfamily we chose to depict the default AnGST output.

**Mobile element analysis.** All genes within the 9CS106 and FF50 genomes were annotated using the svr_assign_using_figfams script from RAST. Regions containing Aly or Oal genes within were defined as any stretch of DNA where an Aly or Oal was no farther than 5,000 bp from the next Aly or Oal. A hypergeometric test as implemented in Python was used to compare the occurrence of genes annotated as mobile element proteins, integrases or transposases within these regions to the occurrence of these genes across the whole genome. GC content of these regions was tested in the same way, with the added restriction of only comparing each region's GC content to the GC content of the chromosome or extrachrosomal element where it was located.

**Validation of copy-number estimation of alginate lyase genes.** We took quality-trimmed sequence reads from 15 strains (ZF29, 9ZC13, 1S175, FF50, 5S101, 1S128, 5F306, 5S186, 1S165, 9ZB36, 12E03, 1F267, 1S45, 5F59 and 9CS106) and searched them against our database of all PL7 domains with UBLAST with default parameters (http://drive5.com/usearch). We then calculated the percentage of all sequenced bases that hit at least one PL7 domain. This quantity correlated well with the predicted number of PL7 genes we obtained from the assembled draft genomes indicating that it is unlikely that unassembled genome regions contain a significant number of new PL7s (Supplementary Fig. 4).

**Size fractionation of alginate.** To prepare alginate of different molecular weight for growth experiments, low viscosity sodium alginate (Sigma #A2158) was size fractionated with chemical and enzymatic approaches. For the production of homopolymeric blocks of alginate with a Dp of ∼Dp20, the chemical method adapted from Haug et al.[40] was used. Briefly, the alginate was heated to 100 °C in 0.3 M HCl for 20 min. After 20 min, the insoluble material was collected by gravity filtration through cotton cloth. The soluble material (containing the heteropolymeric alginate fraction) was discarded. The insoluble material was dissolved in fresh 0.3 M HCl, and hydrolysis was continued at 100 °C under stirring for 20 h. Insoluble material was collected by filtration and suspended in water to which dilute NaOH was added until the solution cleared. The solution was dialyzed against MilliQ water, after which the solution was adjusted to ∼0.5% sodium alginate by addition of water and NaCl to a final concentration of 0.1 M. The alginate was fractionated by addition of diluted HCl to the alginate solution until the pH reached 2.85, inducing separation into a precipitate and supernatant fractions enriched in mannuronate and guluronate, respectively. Both fractions were neutralized with NaOH to obtain a pH of 7 followed by dialysis against MilliQ water with dialysis tubing (MWCO, 1 kDa). Following dialysis, the guluronate- and mannuronate-enriched alginate fractions were precipitated with ethanol, dried at 60 °C, and re-dissolved in MilliQ water. The solutions were frozen at −80 °C and lyophilized to obtain fine powder. Unless otherwise indicated, all steps were carried out at 20 °C.

To prepare oligosaccharides with a Dp of Dp ∼3–4 the mannuronate and guluronate enriched fractions were dissolved to obtain a concentration of 0.5% (w/v) in 100 ml of 10 mM 3-(N-morpholino)propanesulfonic acid (MOPS) buffer pH 7.5 and 2 mM CaCl₂. This solution was amended with alginate lyase (Aly) from Flavobacterium sp. [4.2.2.3] (Sigma, A1603) at a concentration of 2.5 µg ml⁻¹ (w/v), which cleaves alginate, without preference for M or G blocks into oligosaccharides with a Dp ∼3–4 as major products[41,42]. The solution was immediately sterile filtered through a 0.2 µm Sterivex filter and the first 20 ml of the filtrate were discarded. The final filtrate was incubated for 24 h at 20 °C under stirring at 200 r.p.m. under sterile conditions. Growth of contaminating bacteria was tested by examination of subsamples stained with SYBR Green I (Life Technologies) under an epifluorescent microscope. Completeness of digestion was assessed by thin layer chromatography in a solvent system of 1-butanol–acetic acid–water (2:1:1, vol/vol). The thin layer chromatography plates were sprayed with 10% sulfuric acid in ethanol and heated at 100 °C until products became visible. The lysis products of the digestion were compared to standard oligosaccharides of mannuronate and guluronate with an average of ∼Dp3 obtained from Elicityl. Only completely digested preparations, as judged by complete conversion into smaller oligosaccharides (∼Dp3–4) was used for further experiments. Completeness of degradation with Aly was further confirmed by repeated addition of the enzyme to the oligo preparation and by measuring OD at 235 nm to test for additional lysis. Enzymes were removed by pressure filtration through a 10 kDa MWCO Millipore membrane (Millipore) that was previously extensively washed with MilliQ water. The filtrate was again sterile filtered through a 0.2 µm Sterivex filter and the first 20 ml of the filtrate were discarded. The final filtrate was frozen and stored at −20 °C until further use.

**Secreted enzyme screenings.** To test for bacterial ability to secrete alginate lyase enzymes (Alys) into the environment, we used the plate assay described in Gacesa and Wusteman[43]. Cultures were grown for 36 h in Marine Broth 2216 (DIFCO), and 2 µl of each culture was spotted with a 96-well solid pin replicator onto agarose plates made with Marine Broth 2216 (Difco) plus 0.25% of low viscosity sodium alginate (A2158, Sigma). After 24 h of incubation at 20 °C, the colonies were

imaged and subsequently removed by scraping. To image the secreted enzyme activity, the plates were rinsed two times for ten minutes with MilliQ water at 20 °C to remove residual cells. After this washing step the plates were incubated with 50 ml of 10% cetylpyridinium chloride (Sigma) solution for 20 min, while gently shaking at 20 °C. The plates were washed twice with MilliQ water for 20 min to remove unbound cetylpyridinium chloride and to increase contrast. Secreted enzyme activity became visible by a dark halo on an opaque background. The area of the colony and the area of the corresponding alginate lyase activity halo was measured with ImageJ.

**RT-qPCR of Vibrio alginate lyase and oligoalginate lyase genes.** Selected Vibrio isolates were grown in triplicate with glucose as the sole carbon source until the cells reached mid-log phase. Subsamples were taken from the cultures and immediately preserved using RNAprotect (Qiagen) to determine the baseline expression in absence of alginate. Cultures were then spun down, washed, and re-dissolved into minimal media with alginate oligosaccharides as the only carbon source. The cultures were allowed to grow for another two hours before samples were again taken and preserved. Total RNA of all samples was isolated using the RNeasy kit (Qiagen) and treated using the TURBO DNA-free Kit (Life Technologies) to remove all contaminating DNA. RNA purity was confirmed by showing DNA was not detectable in any sample after 35–40 rounds of PCR. First-strand cDNA synthesis was carried out using the SuperScript III Reverse Transcriptase kit (Invitrogen). All reverse transcription reactions were carried out using 100–150 ng of total RNA per reaction and random hexamer primers. Transcript measurement of all alginate lyase genes were performed on a CFX96 Real-Time PCR detection system (Bio-Rad) using SYBR select master mix (Life Technologies) and primers listed in Supplementary Table 3. Gene expression analysis was performed using REST-2009 software (Qiagen) with RpoD and GyrB as reference genes.

**Bulk enzymatic activity assay of Vibrio alginate lyases.** Selected Vibrio strains were grown for 48 h in 96 deep-well blocks with Marine Broth 2216 (DIFCO) with or without 0.25% (w/v) alginate oligosaccharides. The OD 600 of the cultures was taken, and the cultures were centrifuged at 2,000 r.p.m. for 20 min at 4 °C. The supernatant was discarded, and the pellets were lysed in Bugbuster master mix (Merck) for 30 min with shaking. The activity assays were carried out in an activity buffer (10 mM KPO₄, 200 mM NaCl, 200 mM KCl, 2 mM CaCl₂, 0.01 % sodium azide, pH = 7.5) containing 0.1% alginate polysaccharide (Sigma #A2158). For the activity curves, 10 µl of bacterial lysate was incubated with 200 µl of the activity buffer at 27 °C. The absorption of the solution at 235 nm was used to quantify polysaccharide degradation by increased absorption of the new double bond in the non-reducing end sugar. A measurement was taken every minute up to 30 min. The alginate degrading ability of each strain was determined by dividing the slope of the linear part of its activity curve by its OD 600 measurement.

**Growth assays.** To determine growth rates on different types of alginate substrates, Vibrio isolates were grown in triplicate with alginate polysaccharide, fractionated mannuronate (Dp ∼ 20 or ∼ 5) or guluronate (Dp ∼ 20 or ∼ 5), or glucose as a control. All substrates were dissolved at 0.1% (w/v) concentration in the minimal medium based on Thein et al.[44]. The different substrates were inoculated (1:100) with vibrios grown in Marine Broth 2216 (DIFCO) for 36 h. After inoculation, OD 600 measurements were taken at every hour for the first 24 h, every 1.5–3 h for the following 24 h, and also at 56- and 76-h time points. From the measured growth trajectories, the exponential growth rate was estimated (for a single replicate experiment of a given strain in a particular condition) by fitting the data to a logistic growth model via nonlinear regression. In most cases, this procedure provided a reasonable fit to the experimental data. However, notable exceptions were observed; in particular, growth trajectories that reached a peak and then declined in optical density following exponential growth (due to cell clumping or cell death) yielded unreasonable parameter estimates when fit to a simple logistic model. In these cases, only the data points within the exponential growth phase were fit to an exponential growth model, rather than to a logistic model, with a similar nonlinear regression approach. All fits were manually inspected for quality, and parameter estimates for a given strain in a particular growth condition are average values from three technical replicates.

**Enzyme localization experiments.** To localize alginate lyase activity in different cell compartments, we measured alginate lyase activity in intracellular, membrane-bound and extracellular proteomes (after Method I in Thein et al.[44]). Strains were grown for 24 h at 20 °C in Marine Broth 2216 (DIFCO) and cells pelleted from 1 ml subsamples by centrifugation at 3,000g for 5 min. The pellet was washed with sterile filtered artificial seawater (Sea Salts, Sigma) and washed cells were added to 100 ml of minimal medium[45] with 0.25% of alginate (low viscosity, Sigma) as sole carbon source. The cells were grown by shaking culture flasks at 200 r.p.m. at 20 °C for 24 h, or until they reached an OD of 0.9 measured at 600 nm. After growth, an anti-protease tablet (complete, Roche Diagnostics) was added to each culture, dissolved, and the cells were placed on ice. All following steps were carried out on ice or at 4 °C unless otherwise stated. Twenty-five micolitre of cell culture were pelleted. The supernatant was filtered through 0.2-µm filter membranes to obtain cell-free supernatants. The cell-free supernatants of each culture were concentrated ∼10–20

fold in a centrifugal concentrator (Vivaspin) with a 10-kDa cutoff. The cell pellets were suspended in 500 µl of a buffer containing 0.2 M Tris-HCl (pH 8), 1 M sucrose, 1 mM ethylenediaminetetraacetic acid (EDTA) to which 100 µl of lysozyme (5 mg ml$^{-1}$ in MilliQ water, Sigma) was added. The cell suspension was vortexed and incubated for 5 min at 20 °C after which 2 ml of MilliQ water was added. The cells were lysed by adding 3 ml of a solution of 50 mM Tris-HCl (pH 8), 2% (w/v) Triton X-100, 10 mM MgCl$_2$ and 50 µl of DNaseI (Applichem, 1 mg ml$^{-1}$ in MilliQ water). The suspension was gently mixed and stored on ice until it cleared. The lysed cells were ultra-centrifuged at 40,000g for 30 min at 4 °C to pellet the outer membrane fraction. The supernatant was stored on ice. The pellet was washed in 750 µl of the lysis buffer with no added DNase. The pellet was ultra-centrifuged at 40,000g for 30 min at 4 °C. After centrifugation the pellet was washed three times with 500 µl MilliQ water before it was stored on ice until the activity measurements were carried out.

For activity assays the outer membranes were suspended in 250 µl of 20 mM Tris (pH 7.5), 0.1 % Tween 20 and 5 mM dithiothreithol (DTT). The activity measurements were carried out in a buffer containing 50 mM Tris (pH 7.5), 2 mM calcium chloride, 0.1 mM alginate and 0.5 M sodium chloride. 20 µl of each sample (concentrated supernatant, cell lysate and the membrane fraction) was added to 180 µl of activity buffer. The increase in absorbance at 235 nm was measured for 30 min with an absorbance reading every minute in Costar UV transparent microtiter plates (Corning #3635) at 25 °C.

**Sampling of algal detritus and zooplankton.** Algal detritus and zooplankton were collected from Plum Island Sound Estuary, Ipswich, MA in the spring and fall of 2007 as previously described[8,9]. Briefly, algal detritus particles and zooplankton were collected by filtering one hundred liters of seawater through a 64 µm mesh net. Eight replicate 100 l samples were collected in each season, two samples per day. Samples were rinsed three times with sterile seawater, washed into a 50 ml conical tube, and kept at ambient temperature in the dark until processing ~2 h later. Algal derived particles, as well as living and dead zooplankton, were picked from each 100 l concentrate. All collected algal particles and zooplankton were washed three times with sterile seawater, after which they were homogenized in a tissue grinder. Subsequently, these lysates were diluted in sterile seawater and filtered onto 0.2 µm filters (Pall).

**Bacterial isolation and gene sequencing.** Homogenates of algal particles and zooplankton were plated on TCBS media (BD Difco TCBS with 1% NaCl added) to isolate Vibrio strains, as previously described[8,9]. After colonies were allowed to grow, they were re-streaked three times, alternating between 1% Tryptic Soy Broth (TSB) media (BD Bacto with 2% NaCl added) and marine TCBS media. For identification and assignment to previously identified populations, the 16S rRNA gene and three protein-coding genes (hsp60, mdh and adk) were sequenced as previously described [8].

**Data availability.** Bacterial genomes that support the findings of this study have been deposited in GenBank with the accession numbers presented in Supplementary Table 2. Vibrio hsp60, adk, and mdh sequences have been deposited in GenBank under accession nos. GQ988782 to GQ990534 (hsp60), GQ990535 to GQ992287 (adk), GQ992288 to GQ994040 (mdh). All other relevant data supporting the findings of this study are available from the corresponding author upon request.

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

## Acknowledgements

This work was supported by the U.S. Department of Energy (DE-SC0008743) to M.F.P. and E.J.A. J.-H.H., M.S.D., and P.A. were partially supported by fellowships from the Human Frontier Science Program, the Department of Defense through a National Defense Science and Engineering Graduate (NDSEG) Fellowship, and the National Science Foundation through the Graduate Research Fellowship Program (GRFP), respectively.

## Author contributions

J-H.H., M.S.D., A.H., C.H.C., M.F.P. and E.J.A. designed experiments. P.A., M.S.D., J.-H.H., A.H., S.T. carried out genomic comparisons and phylogenetic reconciliation. S.P.P. did the environmental sampling and populations structure analysis. All authors contributed to writing and editing the manuscript.

## Additional information

**Competing financial interests:** The authors declare no competing financial interests.

