## [Peer Review File · Nature Communications]

Reviewers' comments:

Reviewer #1 (Remarks to the Author):

Polz and colleagues present an interesting case study of metabolic specialization in a marine microbial clade. They find three distinct groups of isolates that specialize in utilizing algae-derived polysaccharides at different states of enzymatic degradation. Based on genomic, phenotypic and ecological data, they ascribe these specializations to adaptive radiation mediated by horizontal gene transfer. Overall, the findings nicely demonstrate microbial metabolic specialization, a natural cross-feeding community resulting from this, and provide insights into possible evolutionary paths. The main conclusions, however, need further explanation/supporting data/analysis as described below.

Major comments:

1. Horizontal gene transfer (HGT) hypothesis: Besides harboring alginate utilization genes, do the identified genomic islands show other signals of HGT like G+C signature, repeat structures, or mobile elements?
2. Gene dosage vs regulation: i) Correlation shown in figure 2 is basically only due to the highest copy number group. ii) Considering that the alginate lyases are present in the genomic island and placed closely together, their expression might be regulated by the same promoter. Besides, since both activity assay and rt-qPCR do not include a negative control, it is difficult to conclude that the increased activity is not because of change in regulation but because of gene acquisition by HGT.
3. Ecological model of metabolic specialization (Figure 5): The model raises a question of stability of the proposed system where the "harvesters" and the "scavengers" are in essence "cheaters" leaving off the work done by the "pioneers". This could possibly operate as a food-chain but would require data supporting that the population sizes of these different groups fit in such a model.
4. Copy number estimates: Are all the genomes of the described *V. breoganii* subspecies complete? It will be good to control for the possibility that the copy number estimates are biased due to incomplete genomes, esp. for PL7 for which the copy numbers are low (only one or two).

Reviewer #2 (Remarks to the Author):

This is a great study, suggesting that the recently evolved diversification of some co-occurring *Vibrio* populations may have been driven by specialisation on different complexities of alginate molecules. Phylogenetic analyses shows that the presence of particular alginate lyase genes is caused by rapid HGT, and crucially, the presence of particular genes corresponds with growth phenotypes and the environments where the populations are found.

I have a few minor comments.

Fig 3 is a key figure, and it would be useful to indicate on the figure the number and types of genes present in each population.

For fig 3, fig 4 and environmental data in fig 1, it would be useful to formalise associations between gene content and phenotype by carrying out some simple categorical tests of association (i.e. Fisher's exact test).

The suggested model of cross-feeding interaction between the different phenotypes is analogous to Morris's (2012 mbio) black queen hypothesis. It would be worthwhile to make this connection.

The cited experimental adaptive radiation reviews all deal with in vitro studies. Gomez (2013 ecol letts) show an experimental adaptive radiation in soil, and worth citing here.

Reviewer #3 (Remarks to the Author):

This is an extraordinarily important paper in this group's series of excellent studies on bacterial "speciation", finely mapping ecotype to genotype and documenting underlying genetic processes. Absolutely should be published in Nature. I have three larger concerns and three smaller.

1. The word 'species'. I suspect that authors and I might agree that this word means what you want it to mean and that it is OK to talk about "speciation" without defining it. But when I see in Fig, 1 that what are referred to as 'populations' in the text are labeled as distinct 'species', I get a little confused. Most folks think that 'populations' are smaller than, or parts of, 'species'. Maybe they need a little philosophical paragraph where they discuss the unsolvable "species problem" and spell out the rules under which they use these words.

2. The biggest surprise here of course is that these varying sets of paralogous gene families are created by HGT rather than losses, duplications and divergences. The proof against the latter is in the trees and about how good these might be and how strong, statistically, is their claim for HGT. Maybe more statistical words in the text and a few trees in a text (nor supplemental) figure? So much depends on this!

3. Personally I like to think that homologous recombination within "species" is a different sort of thing than HGT, even though it may be one of the processes normally associated with HGT that brings the DNA in so that it can be recombined. There seems not to be much about the genetics here. Are all those green genes in Fig. 1B linked, so that it is just a matter of bringing a new one in and recombining it with the aid of flanking homology, or is each in its own chromosomal location so that some illegitimate recombinational processes were involved? What they call "surprisingly facile incorporation and expression of heterologous genes" would be easier to believe if structures of the relevant resident gene clusters facilitated gene (even "allele") replacement. In a way, what we are looking at seems not unlike the distribution of lactase alleles in human populations associated with

dairy practices. If we looked at the "crown group" of seven "species" as just one "species", would this not be pretty similar?

Smaller issues.

1. Second paragraph of Intro ... might be good to spell out what sort of "genotypic clusters" were "originally identified". Did not do whole genomes back in the day.

1. Not sure why or how the "surprisingly facile incorporation and expression of heterologous genes" is "reminiscent of molecular cloning and exemplifies why the process works in the laboratory".

2. In Fig. 1B, it would be good to tell us which are Aly and which Oals genes.

Reviewer responses NCOMMS-16-04725

We would like to thank all reviewers for the thoughtful reading of the manuscript. We feel that the manuscript has been improved considerably by addressing all the comments.

Additional changes:

Abstract: We put the abstract in present tense to comply with Nature formatting.

Results: We inserted subheadings.

Figure 1C: Habitat information for *Vibrio lentus* and *Vibrio tasmaniensis* was incorrectly attributed to *Vibrio cyclitrophicus*. Additionally, habitat information was distorted by resizing and has been corrected.

Figure 1A and D, Supplementary Fig. 1: Renamed *Aliivibrio logei* to *Aliivibrio logei-like* and *Aliivibrio salmonicida* to *Aliivibrio logei* based on refined placement of type strains.

Reviewer #1:

Polz and colleagues present an interesting case study of metabolic specialization in a marine microbial clade. They find three distinct groups of isolates that specialize in utilizing algae-derived polysaccharides at different states of enzymatic degradation. Based on genomic, phenotypic and ecological data, they ascribe these specializations to adaptive radiation mediated by horizontal gene transfer. Overall, the findings nicely demonstrate microbial metabolic specialization, a natural cross-feeding community resulting from this, and provide insights into possible evolutionary paths. The main conclusions, however, need further explanation/supporting data/analysis as described below.

Major comments:

1. Horizontal gene transfer (HGT) hypothesis: Besides harboring alginate utilization genes, do the identified genomic islands show other signals of HGT like G+C signature, repeat structures, or mobile elements?

In addition to the phylogenetic evidence presented in the original manuscript, we have three additional lines of evidence that point to the mobility of oligoalginate lyase (Oal) and alginate lyase (Aly) genes within the Vibrionaceae. These stem from analysis of newly available closed genomes from a member of *Vibrio breoganii* (FF50) and a nearly closed genome of a member of the crown group population *Vibrio* sp. F13 (9CS106). We identified 8 distinct regions within FF50 and 4 regions with 9CS106 that contained Alys or Oals.

1. These regions are distributed throughout the genome in both FF50 and 9CS106. In FF50, there are 3 regions on chromosome I, 8 regions on chromosome II, and 2 regions on a putative extrachromosomal element (ECE). In 9CS106, there are 3 regions on chromosome I and 1 region on chromosome II. Moreover, the Alys and Oals in FF50

occurring on chromosome II are orthologs of the genes that appear on chromosome I in 9CS106.

2. The regions around oligoalginate lyases and alginate lyases are significantly enriched in genes predicted to be integrases, transposases, and part of other mobile elements. These predicted mobile genes compose 2.5% of all open reading frames (ORFs) across the genome. In contrast these genes make up 5.9% of all ORFs within regions containing alginate and oligoalginate lyases. This difference is significant according to a hypergeometric one-tailed test ($p = 0.0019$).
3. The GC content of some of these regions is significantly lower than chromosome-wide GC content. There are 8 regions within FF50 containing Alys and Oals. Of these 8 regions, 4 have a significantly decreased GC content (Bonferonni-corrected $p < 0.01$). Similarly, Aly or Oal-containing regions were identified in 9CS106. Of these, one has significantly reduced GC content (Bonferonni-corrected $p = 2.27e-6$).

We have included this information in lines 128ff of the main text: “Moreover, Alys and Oals are distributed across multiple regions on chromosome 1, chromosome 2, and a putative extrachromosomal element in one *V. breoganii* (FF50) and one *Vibrio* sp. F13 (9CS106) isolate with nearly closed genomes. These regions are significantly enriched in genes annotated as mobile elements, transposases, and integrases (hypergeometric test, $p = 0.0019$). Some of these regions also display significantly decreased GC content consistent with the recent introduction of foreign DNA (Supplementary Table 1).”

2. Gene dosage vs regulation: i) Correlation shown in figure 2 is basically only due to the highest copy number group.

This does not seem to be the case. If data from the highest copy number group (# of copies > 10) are excluded, the correlation is 0.81, compared to 0.87 when the highest copy number group is included.

- ii) Considering that the alginate lyases are present in the genomic island and place closely each other, their expression might be regulated by the same promoter. Besides, since both activity assay and rt-qPCR do not include a negative control, it is difficult to conclude that the increased activity is not because of change in regulation but because of gene acquisition by HGT.

Although some of the alginate lyases are present in the genomic island shown in Supplementary Figure 2, the genomes contain additional alginate lyases outside of this island. Those outside alginate lyases are frequently not localized close to each other and they are not part of obvious operons for alginate catabolism. Because they are positioned in distant locations of the genomes this limits the possibility for operon like expression of the entire set of alginate lyases.

To clarify the issue and in agreement with the reviewer’s concern, we have rephrased the sentence: “Hence, increased activity rapidly evolved by gene acquisition and expression rather than evolution of regulation.” to: “Because many of the genes are distributed across different

regions in the genome and hence not co-regulated, increased activity rapidly evolved, in large part by gene acquisition and expression.” (line 162)

3. Ecological model of metabolic specialization (Figure 5): The model raises a question of stability of the proposed system where the "harvesters" and the "scavengers" are in essence "cheaters" leaving off the work done by the "pioneers". This could possibly operate as a food-chain but would require data supporting that the population sizes of these different groups fit in such a model.

We agree that this point should be clarified. Thus, we have adjusted the text (second paragraph in discussion) to discuss the mechanisms by which pioneers, harvesters, and scavengers may coexist. We have to rely on literature since we cannot extract precise parameters for our populations since we do not know the exact population sizes and frequencies in situ and we do not know the detailed environmental factors that drive the population dynamics in the environment at relevant microscales. Steve Allison modeled pioneer and cheater dynamics in silico, finding that spatial structure and diffusion regulate the success of pioneers with extracellular enzymes versus cheaters without such enzymes. Depending on changing physical parameters he calculated very different outcomes of abundances of either groups (10.1111/j.1461-0248.2005.00756.x). We have now included this work and other work addressing this problem in the discussion:

Line 263ff: “In theory, there are many mechanisms that might support the coexistence of these three ecological strategies in nature. For pioneers and scavengers, several mechanisms may support their coexistence, including spatial structure [Allison, 2005; Allen, et al., 2013], asymmetric access to nutrients [Gore, et al., 2009], and others [Celiker and Gore, 2013]. For harvesters, their lack of broadcast enzymes leads to a growth detriment (through long lag phases during growth on high molecular weight alginate), but also makes them less likely to share their enzymatic degradation products. Thus, harvester populations may not be as prone to invasion by scavengers – as has been recently described for select human gut *Bacteroidetes*²⁸—thereby allowing them to coexist with pioneers and scavengers. Finally, even different pioneer populations (*V. breoganii* and *Vibrio* sp. F13) are further ecologically differentiated by enzymatic activity levels, stemming from distinct pathway architectures (Fig. 2 and Supplementary Fig. 4), which may allow for their coexistence. Nonetheless, Interactions may lead to fluctuations in populations and further work will be required to determine how stably pioneers, scavengers, and harvesters can coexist in the wild.”

4. Copy number estimates: Are all the genomes of the described *V. breoganii* subspecies complete? It will be good to control for the possibility that the copy number estimates are biased due to incomplete genomes, esp. for PL7 for which the copy numbers are low (only one or two).

As described above, we have completed one *V. breoganii* genome since submission of the manuscript and have one nearly-complete *Vibrio* sp. F13 genome. Overall, we believe that our inferred lyase counts are accurate despite having mostly genomes that are not formally closed for our collection. Our confidence is based on two pieces of evidence in response to the reviewer's concern:

1. The PacBio-based genome sequences of 9CS106 and FF50 indicated that our counts based on draft genomes were accurate. Applying our same HMMER search approach to the new 9CS106 genome yielded no additional Alys or Olys. One additional PL7 (Aly) was found in the *V. breoganii* genome FF50. The total number of inferred PL7s in the new FF50 genome matches maximum number of PL7s we observed among all draft *V. breoganii* genomes.
2. We took quality-trimmed sequence reads from 15 strains (ZF29, 9ZC13, 1S175, FF50, 5S101, 1S128, 5F306, 5S186, 1S165, 9ZB36, 12E03, 1F267, 1S45, 5F59, 9CS106) and searched them against our database of all PL7 domains with UBLAST. We then calculated the percentage of all sequenced bases that hit at least one PL7 domain. This quantity correlated well with the predicted number of PL7 genes we obtained from the assembled draft genomes indicating that it is unlikely that unassembled genome regions contain a significant number of new PL7s. We have included this in formation in Supplementary Figure 4, referenced in the text in line 158 and have added a section to the methods on “validation of copy number estimation of alginate lyase genes” (line 437ff).

Reviewer #2:

This is a great study, suggesting that the recently evolved diversification of some co-occurring *Vibrio* populations may have been driven by specialisation on different complexities of alginate molecules. Phylogenetic analyses shows that the presence of particular alginate lyase genes is caused by rapid HGT, and crucially, the presence of particular genes corresponds with growth phenotypes and the environments where the populations are found.

I have a few minor comments.

1. Fig 3 is a key figure, and it would be useful to indicate on the figure the number and types of genes present in each population.

We agree that presenting this information would be useful. However, indicating it on the figure is difficult to do clearly, since there is variability in gene copy number across different strains within a population. Instead, we have added a second panel to Figure 3, which indicates the number and types of alginate lyases present in individual strains within each population.

2. For fig 3, fig 4 and environmental data in fig 1, it would be useful to formalise associations between gene content and phenotype by carrying out some simple categorical tests of association (i.e. Fisher's exact test).

Thank you for the suggestion. We have added statistical tests of these associations throughout the main text (lines 175ff, 232).

3. The suggested model of cross-feeding interaction between the different phenotypes is analogous to Morris's (2012 *mbio*) black queen hypothesis. It would be worthwhile to make this connection.

The paper by Morris et al. is very important for explaining the structure of microbial communities but we think it does not fully apply here. The authors suggested that mutualistic interactions such as reciprocal cross feeding may derive from gene loss of key functions in one partner. These functions must be compensated by the other partner, which may also lose other genes which are then compensated again by the other partner. This may lead over time to a form of dependency. We are not sure that the Black Queen Hypothesis applies here because we do not know if the populations described here are reciprocally cross feeding.

4. The cited experimental adaptive radiation reviews all deal with in vitro studies. Gomez (2013 *ecol letts*) show an experimental adaptive radiation in soil, and worth citing here.

Thank you for bringing this study to our attention; we agree that it is relevant. We have now cited this work at **line 59** of the main text.

Reviewer #3:

This is an extraordinarily important paper in this group's series of excellent studies on bacterial "speciation", finely mapping ecotype to genotype and documenting underlying genetic processes. Absolutely should be published in Nature. I have three larger concerns and three smaller.

1. The word 'species'. I suspect that authors and I might agree that this word means what you want it to mean and that it is OK to talk about "speciation" without defining it. But when I see in Fig. 1 that what are referred to as 'populations' in the text are labeled as distinct 'species', I get a little confused. Most folks think that 'populations' are smaller than, or parts of, 'species'. Maybe they need a little philosophical paragraph where they discuss the unsolvable "species problem" and spell out the rules under which they use these words.

We realize that our use of population and species can cause confusion and have tried to clarify the issue by rewriting the second paragraph of the introduction. Our rationale is as follows: All our isolates stem from the same water samples so that we believe they can fulfill the criterion of coexistence necessary for populations within species. In many cases, these populations fall within the diversity of previously described "species" so that we refer to them using these species names. We have tried to clarify this in the text by replacing species with clades (line 71) and modifying the description of the populations to include our rationale for naming them (line 78): "However, because our sampling scheme considers only bacteria co-existing in small-scale microhabitats, we designate them as populations to which we assign species names if a previously described type strain falls within the genotypic cluster identified as a distinct population."

We do appreciate the suggestion of a discussion of the species problem but we fear that this might distract from the essence of the paper. Our focus is on the description of evolutionary processes rather than the nature of species and we do not want to complicate the matter where it does not seem necessary.

2. The biggest surprise here of course is that these varying sets of paralogous gene families are created by HGT rather than losses, duplications and divergences. The proof against the latter is in the trees and about how good these might be and how strong, statistically, is their claim for HGT. Maybe more statistical words in the text and a few trees in a text (nor supplemental) figure? So much depends on this!

This is a very important point and we have engaged in an extensive analysis, which shows that the basic conclusion in the paper is robust. The critical point is that the exact outcome of the reconciliation depends on how subfamilies are defined. Our additional analysis (detailed below)

shows that even if we define subfamilies more conservatively in a couple of borderline cases, there are no additional inferred ancestral duplications.

In the original manuscript, we defined subfamilies of each Aly and Oal as Vibrionaceae-specific clades that are divided by non-Vibrionaceae outgroups. Some of these clades were, however, so closely related that two alternative scenarios are indeed possible: either horizontal acquisition of two different (albeit closely related) subfamilies or vertical evolution within the Vibrionaceae and transfer to the outgroup. We therefore tested the robustness of these initial subfamilies to arrive at the most conservative evolutionary scenario using the following methods:

- 1) Bootstrap support values
- 2) Testing the inferred ML topology against a more conservative topology (i.e., a topology that grouped Vibrionaceae genes into a single clade) with the approximately unbiased test (AU test).
- 3) Comparing branch lengths within a given subfamily phylogeny to branch lengths within a related subfamily with an origin supported by criteria 1 and 2.

We performed these analyses on the PL17, PL6, and PL7 phylogenies. PL15 was not included in this analysis as only one PL15 subfamily was found. The results of these analyses have been added to the manuscript as an extensive description in the methods (“statistical support of alginate lyase subfamilies, line 357ff) and we refer to this analysis in the results (line 120). We have also performed a new reconciliation analysis shown in Fig. 1, which, although different in some details, confirms the original conclusion. As suggested by the reviewer, we have also added a phylogenetic tree documenting an important part of our analyses (Fig S10). Because the trees are generally extremely large, we find it impossible to add more figures in the main text but we note that the supplementary already contains 4 tree figures documenting the phylogenetic structure of lyase families across all Vibrionaceae.

PL6:

Originally two subfamilies were inferred. Subfamily 1 was well-supported by bootstraps (97) while subfamily 2 was not (42). We tested the alternative hypothesis that subfamilies 1 and 2 were a monophyletic group and found that this topology was significantly different from the inferred ML topology ($p = 2 \times 10^{-4}$). **We were able to reject the alternative hypothesis and are confident that there are at least two independent subfamilies of PL6 within the Vibrionaceae.**

PL7:

Originally 14 subfamilies were inferred. Subfamilies 1 and 14 were supported by high bootstraps (100 and 90 respectively). Subfamily 13 was most closely related to subfamily 14 and therefore we tested the alternative hypothesis that these two subfamilies formed a

monophyletic clade. This alternative topology was significantly different from the ML topology ($p = 0.017$) and therefore we were able to reject that alternative hypothesis.

Subfamilies 3-12 form a large clade along with a number of non-*Vibrio* members. Because of this, testing an alternative hypothesis was not appropriate given the number of possible alternatives available. We took advantage of the fact that subfamily 14 originated in the ancestor of *Aliivibrio* and *Vibrio*, making it the oldest of all Vibrionaceae PL7s tested. The longest root-to-leaf branch length in this subfamily is indicative of the maximum amount of divergence one would expect within any Vibrionaceae-specific PL7 subfamily. Using this value (0.66 substitutions/site) as a cutoff, we were able to recover subfamilies 3-8 (see figure below, also included as Supplementary Figure 10). However, subfamilies 9-12 were merged into a single subfamily. **We performed the reconciliation analysis with this larger subfamily and transfer still dominates the evolution of this larger subfamily.**

The distance between subfamily 2 and all other subfamilies greatly exceeds the cutoff described above and therefore no changes were made to its reconciliation in our revised analysis.

PL17:

Originally five subfamilies were inferred. However, bootstrap support for all of these subfamilies was low (all values < 60). We tested the alternative hypothesis that subfamilies 1, 2, and 3 were a monophyletic group with the AU test. This topology was

not significantly different from the inferred ML topology ($p = 0.339$). Performing the same test on subfamilies 4 and 5 yielded the same result ($p = 0.118$). However, the alternative topology with subfamilies 1, 2, 3, 4, and 5 as a monophyletic clade was significantly different from the ML topology ($p = 7 \times 10^{-60}$) and therefore we were able to reject that alternative hypothesis. **Therefore, we performed the reconciliation again with 2 subfamilies instead of 5 subfamilies (Subfamily 1 + 2 + 3 and Subfamily 4 + 5). Ancestral duplication is still not inferred as part of the reconciliation and indeed, even more complex transfer dynamics emerge within the crown group.**

Overall, we feel that these results are more conservative now and strengthen the conclusion that transfer rather than duplication is responsible for the large variation in copy numbers in these lyase genes. We thank the reviewer for the suggestions leading to this analysis.

3. Personally I like to think that homologous recombination within "species" is a different sort of thing than HGT, even though it may be one of the processes normally associated with HGT that brings the DNA in so that it can be recombined. There seems not to be much about the genetics here. Are all those green genes in Fig. 1B linked, so that it is just a matter of bringing a new one in and recombining it with the aid of flanking homology, or is each in its own chromosomal location so that some illegitimate recombinational processes were involved? What they call "surprisingly facile incorporation and expression of heterologous genes" would be easier to believe if structures of the relevant resident gene clusters facilitated gene (even "allele") replacement. In a way, what we are looking at seems not unlike the distribution of lactase alleles in human populations associated with dairy practices. If we looked at the "crown group" of seven "species" as just one "species", would this not be pretty similar?

We tend to agree that homologous recombination within species is different from non-homologous recombination. As detailed in the response to Reviewer 1 (comment #1), we did find much evidence for "illegitimate recombinational processes" (i.e., mobile elements and bacteriophage) mediating transfer of these lyase genes. However, we believe that homologous recombination also played a role in the pathway's history. We've found at least one example that suggests flanking homology may have mediated transfer of a PL7 alginate lyase in *V. splendidus* strain 1S124. The figure below shows two genomic regions of 2 *V. splendidus* strains, one of which contains an alginate lyase (1S124) and one that does not (12B01). This particular PL7 is only present in 2 other *V. splendidus* strains (13B01 and FF6) and occurs in the same genomic context in both strains. Additionally, these three strains do not form a monophyletic clade within *V. splendidus*. *V. splendidus* strains such as 12B01 that lack this lyase possess a matching region with only PL7 missing. Thus, it is plausible that after an initial acquisition of this PL7 within one *V. splendidus* strain, it spread to other strains through homologous recombination of the flanking regions.

We appreciate the comment whether we are looking at populations within a single species vs. populations of different species. We do believe these populations are on different evolutionary trajectories. We have previously characterized the population-specific spread of adaptive genes (Shapiro et al. 2012) and ecological tradeoffs consistent with niche separation (Yawata et al. 2014). Consistent with this, we see differential environmental distribution across the crown group. Finally, our recent analysis of gene flow (unpublished) shows that the populations of the crown group act as gene flow units albeit there is still considerable genetic exchange due to the close relationships. Hence we believe that these populations have and are undergoing speciation beyond what might be considered within species differentiation.

Smaller issues.

1. Second paragraph of Intro ... might be good to spell out what sort of "genotypic clusters" were "originally identified".

We added in **line 73**: "...in protein-coding marker genes..."

2. Not sure why or how the "surprisingly facile incorporation and expression of heterologous genes" is "reminiscent of molecular cloning and exemplifies why the process works in the laboratory".

We agree that this sentence was not clear and have revised it:

Line 164: This is reminiscent of molecular cloning and exemplifies that the process works in the laboratory because microbes are well adapted for incorporation and expression of heterologous genes.

3. In Fig. 1B, it would be good to tell us which are Aly and which Oals genes.

We have revised the Figure 1 legend and Figure 1B to include this information.

REVIEWERS' COMMENTS:

Reviewer #1 (Remarks to the Author):

The revised manuscript addresses all my comments and is accordingly much improved.

Reviewer #2 (Remarks to the Author):

I'm happy with the authors' responses

Reviewer #3 (Remarks to the Author):

I'm happy with all these revisions and support publication